# Decision analysis of IoT-based big data analytics in smart cities of Saudi Arabia

Wael Alosaimi[1], Abdullah Alharbi[1], Hashem Alyami[2], Bader Alouffi[2], Ahmed Almulihi[2], Masood Ahmad[3], Mohd Nadeem[4] and Raees Ahmad Khan[5]

[1] Department of Information Technology, College of Computers and Information Technology, Taif University, Taif, Saudi Arabia
[2] Department of Computer Science, College of Computers and Information Technology, Taif University, Taif, Saudi Arabia
[3] Department of Computer Engineering & Applications, GLA University, Mathura, India
[4] Department of Computer Science and Engineering, Shri Ramswaroop Memorial University, Barabanki, UP, India
[5] Department of Information Technology, Babasaheb Bhimrao Ambedkar University, Lucknow, UP, India



Corresponding author
Mohd Nadeem,
mohd.nadeem1155@gmail.com

## ABSTRACT

The smart cities are being created to enhance livability for residents by utilizing Internet of Things (IoT) technology to improve urban infrastructure, increase services, and cut costs globally. Saudi Arabia's Vision 2030 aims to transform urban living through Smart Cities (SCs), leveraging IoT-based big data analytics to enhance infrastructure, services, and cost-efficiency. Decision analysis is the most important task in a high volume of changing data, in SCs. This study explores how IoT and big data analytics can optimize decision-making for SC development in Saudi Arabia. The data analytics (DA) enables the gathering, processing, and analysis of enormous amounts of data coming from various IoT-based infrastructures across the nation. IoT-based DA is a fast-emerging domain of Information Technology (IT) infrastructure in SC. The quantitative research employs the Fuzzy Analytic Hierarchy Process (FAHP) to evaluate key factors. The analysis reveals economic growth as the top priority (weight: 0.24), followed by good governance (0.103) and transport (0.095), with economic growth's subfactor (G12) scoring highest at 0.17016. The study proposes a framework for integrating IoT-based big data analytics, examines opportunities and challenges, and offers recommendations for effective implementation, fostering sustainable urban ecosystems. The study also pursues the opportunities and challenges in using IoT-based big data and DA in SCs, and recommends mechanisms for the successful implementation of such initiatives.

## INTRODUCTION

Saudi Arabia is rapidly moving towards the development of Smart Cities (SCs) to improve the quality of life of its citizens. SCs use Internet of Things (IoT)-based technology and data to provide better services and enhance the living standards of their residents. The IoT plays a vital role in improving citizens' life standards. The implementation of SCs requires the collection, analysis and use of the vast amount of data generated by IoT devices (*Tundys &*

*Wiśniewski, 2023*). Big data analytics (BDA) has become instrumental because it enables the processing and analysis of large volumes of data generated by the IoT and enables devices from different sources, leading to better development and providing efficient service delivery (*Abberley et al., 2017*). Strengthening the BDA ensures the analysis of extensive and intricate datasets to extract valuable insights that can improve decision-making processes (*Gayer, Chernyshova & Mamai, 2021*; *Talebkhah et al., 2021*). SCs can provide better services to citizens *via* an IoT monitoring system, promote economic growth, and foster a more live, resilient urban environment with the commitment to empowering citizens and ensuring more holistic lifestyles. The Saudi Arabian government has taken several initiatives to develop SCs across the country by installing an IoT-based data collection system. These initiatives use advanced technologies such as the IoT and BDA (*Liu, 2020*). Thus, the subject in itself becomes an important premise for research with the aim of examining the existing gaps, if any, and citing the prospective endeavors that need to be taken for the optimum use of IoT-based BDA in creating a workable ecosystem for SCs. More specifically, the objective of this research is as follows:

- To investigate the use of the IoT based on BDA in SC projects in Saudi Arabia.
- The impact of decision-making is evaluated *via* the Fuzzy Analytic Hierarchy Process (FAHP).
- To evaluate the weights of the factors associated with big data, they are ranked.
- To examine the impact of data analytics in the SC of Saudi Arabia.
- To map the IoT and big data are in SC projects.

This research employs a quantitative approach and utilizes literature reviews, which are case studies that address this context (*Wu et al., 2018*; *Nguyen, Nguyen & Bui, 2022*). The article begins with an introduction to the IoT and BDA and some relevant points in the case of SC.

In 'Literature Review', the literature on the IoT and BDA in SCs is examined, and a paradigm for their integration is presented. In 'Materials: Smart City Factors', the SC and its factors related to Saudi Arabia are identified, and big data and the DA in the SC are discussed. 'Methods' discusses the materials and methodology, the research framework process, and the decision-making methodology. In 'Results', the rankings of the factors are evaluated, and the results are mentioned and tabulated. In 'Discussion', the results are compared with those of classical and fuzzified approaches. 'Sensitivity analysis and Comparisons' suggests the recommendation of the SC project. In 'Conclusions', the conclusions of the IoT and BDA, DA, and SC initiatives of Saudi Arabia are discussed. The article concludes by highlighting the need for further research on this topic.

## LITERATURE REVIEW

As the world becomes increasingly digitalized, the importance and need for IoT-based devices and their collection of data are called 'IoT-based big data (IBD)'. In the context of managing various services from civic amenities to healthcare, facilitating public conveyance, amongst a host of the other day-to-day transactions of the services in SC, IBD

assumes a pivotal role. The research literature may be utilized to improve the efficacy and efficiency of several SC sectors, including public safety, energy, transportation, and healthcare. For example, it may be used to analyse traffic patterns and improve traffic flow in the transportation sector to ease congestion and enhance commuter experiences (*Nguyen, Nguyen & Bui, 2022*). Similarly, in the energy sector, it can be used to scrutinize energy consumption patterns and optimize energy usage to reduce waste and lower costs. IBD can be utilized in healthcare to analyse healthcare data and provide customized treatments for patients. Moreover, it can help in predictive analysis of the spread of communicable diseases in the susceptible zones of the SC, thus containing contagious infections and saving human lives.

In the public security sector, it can be used to analyse crime data and identify possible hotspots for targeting interventions to improve public safety. Numerous studies have examined the role of BDA in SCs. For example, *Reddy & Mehta (2019)* investigated the application of BDA in intelligent transportation systems. Their study revealed that BDA can optimize traffic flow to reduce congestion and enhance public transportation services. Similarly, *Jurado Pérez & Salvachúa (2021)* investigated the use of BDA in smart healthcare systems. This study revealed that BDA can be used to improve the diagnosis and treatment of diseases.

*Huang & Nazir (2021)* utilized the analytic network process to evaluate SCs for the IoT on the basis of their potential and cases. To meet the immediate demands of the expanding population and promote the growth of the SC, weighing a range of time-sensitive and successful alternatives is essential. Additionally, the expansion of the IoT has spawned other study fields that might aid in the advancement of SCs. In light of the potential use cases for SCs, *Huang & Nazir (2021)* proposed the use of the decision-making approach to evaluate and prioritize various aspects of SC development.

*Yu, Mihai & Srivastava (2021)* presented a complete system that uses a variety of IoT-based smart devices to gather data, including smart homes, vehicle networking, and smart parking. The proposed system leverages the hadoop ecosystem to facilitate its implementation. To assess the system's effectiveness, its throughput and processing time were evaluated. The results indicate that the proposed approach outperforms the existing techniques by 20% to 65% in terms of processing time and by 20% to 60% in terms of obtained throughput (*Yu, Mihai & Srivastava, 2021*). Table 1 presents recent research studies based on data analytics and IBD. The rapid advancement of IBD has transformed smart SC development, enabling data driven decision making to enhance urban liveability. SC initiatives are to diversify the economy and improve citizen's quality of life. This study fills this gap by proposing a tailored IBD framework and using FAHP to rank SC factors, offering a nuanced approach that departs from and improves upon earlier methodologies. The existing literature, while comprehensive, fails to address two key aspects, the application of FAHP to prioritize SC factors in the context of Saudi Arabia's unique socioeconomic landscape, and the integration of a holistic IBD framework that combines real-time and batch analytics for SC decision-making. Previous FAHP based studies focus on non-SC domains, and SC-specific studies using other methods do not adequately

**Table 1 Recent research on IBD and DA.**

| Author | Research Finding |
|---|---|
| Y. Jiang et al. (2023) | The article identifies the impact of crowdsourcing, factors, and different data source of big data in DA (*Jiang et al., 2023*). |
| S. Siddiqui et al. (2023) | Presents the authentication and authorization mechanism in DA security mechanism in SC (*Siddiqui et al., 2023*). |
| A. Khan et al. (2022) | The research provides the comprehensive analysis and challenges on SC. The issues of big data and DA are selected from this (*Khan et al., 2022a*) state of art. |
| O. Samuel et al. (2022) | Used improved sparse neural network in cloud data analysis, blockchain based security mechanism for SC (*Samuel et al., 2022*). We adopted the security concern from this literature. |
| J. Mondschein et al. (2021) | Works on north American SC case study, security and privacy with organizational behavior (*Mondschein, Clark-Ginsberg & Kuehn, 2021*). |
| M. A. Khan et al. (2022) | Highlighted the technical barrier of smart city project for sustainable development (*Khan et al., 2022b*). |
| N. Ianuale et al. (2016) | Highlight the social, technical, economical, and political factors of SC and explain the heterogeneous and diverse data sources (*Ianuale, Schiavon & Capobianco, 2016*). |
| S. A. Shah et al. (2019) | Gives the framework and concept of disaster resilient SC (*Shah et al., 2019*). The IBD analytics with DA process have discussed. |
| M. V. Moreno et al. (2016) | Present the IBD architecture for SC project (*Moreno et al., 2016*). The energy management are discuss in SC. |

address uncertainty or provide a prioritized ranking of factors like economy, governance, and transport.

# MATERIALS: SMART CITY FACTORS

The term SC does not have a universally agreed-upon definition, and there are various definitions in the literature. There is a consensus that information and communication technology (ICT) and the IoT play central roles in realizing a city's smartness (*Yu, Mihai & Srivastava, 2021*; *Pla-Castells et al., 2015*). Depending on context, intelligence might refer to marketing, engineering, or technology, all of which rely heavily on digital artefacts such as sensors, actuators, mobile devices, and IoT applications (*Babar & Arif, 2017*). The idea of SC is to use data and digital technology to improve inhabitants' well-being and decision-making. The implementation of an IoT-based data collection system has attracted the attention of BDA in recent years (*Anisetti et al., 2018*). Approximately 22 billion electronic devices were in use worldwide by the end of 2018, nearly three per person, and one might expect that significant progress would have been made in addressing global challenges (*Nuseir, Mohammed & Aljumah, 2020*). However, despite these IoT-based technological advancements, approximately 10 people still lack access to electricity (*Ilyas, 2021*), poverty-related factors contribute to the death of children every 5 s (*Jin et al., 2014*), and the planet faces a mounting climate crisis (*Jin et al., 2020*).

Improving different domains of a city with advancements in IoT-based systems alone does not necessarily qualify it as an SC. An SC is defined at the city level as a full system of factors, as illustrated in Fig. 1, which takes into account the linkages between the underlying autonomous IoT-based systems and subsystems (*Khan et al., 2022b*). This IBD analysis approach for an SC necessitates the sharing of information across different domains and stresses the distinction between enhancing in a particular city domain and

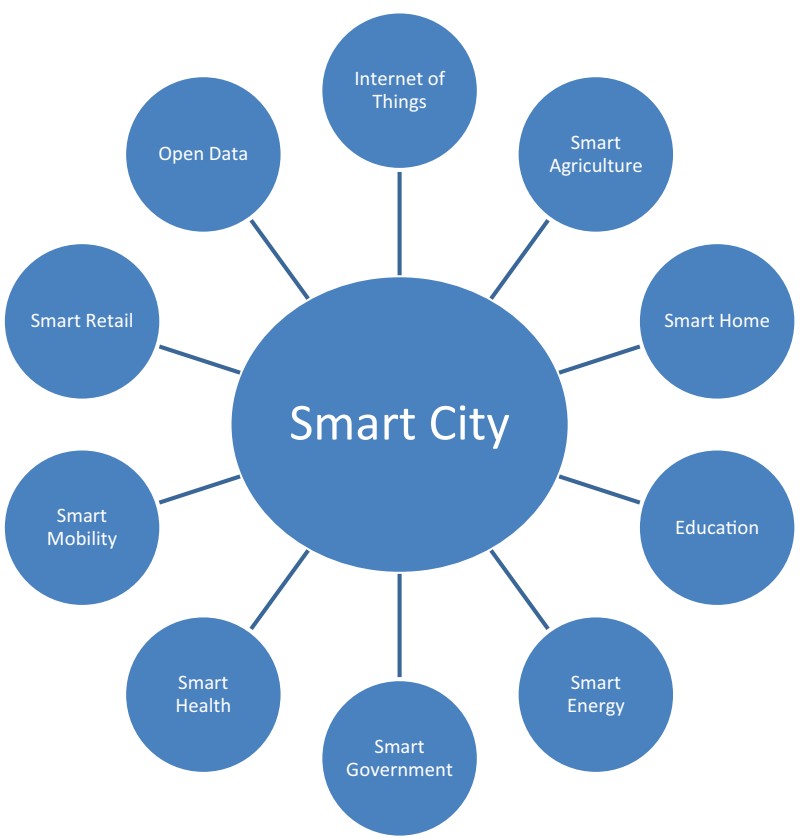

**Figure 1 Smart city concept.**

making the entire city smart. Consequently, evaluating and modelling SCs is a complex process (*Allam et al., 2022*). More aptly, an SC can be defined as an urban area that leverages advanced IoT-based systems to increase the well-being of its inhabitants. The use of an IoT-based system allows for better management of resources, including the economy, transportation and governance, as shown in Fig. 2. Saudi Arabia is implementing steps to minimize its dependency on oil, increase its sources of income, and increase its dependencies on oil, which are causing changes in the country's economy. The ambitious Vision 2030 journey of Saudi Arabia this year indicates a critical turning point (*Alam et al., 2021*; *Alhakami et al., 2023*). The governance, economy and transport infrastructure are modernized under the Vision 2030 plan (*Alhakami et al., 2023*).

## Governance

To assist decision-making, smart governance uses ICTs for data collection, processing, and analysis. This improves the understanding of the status of the city, fosters better communication, improves the delivery of public services, and increases transparency *via* the use of evidence-based policies (*Barns, 2018*). To implement its Vision 2030 plan, several governance challenges must be faced to achieve its ambitious goals. Policy and Infrastructure in State-owned Enterprises.

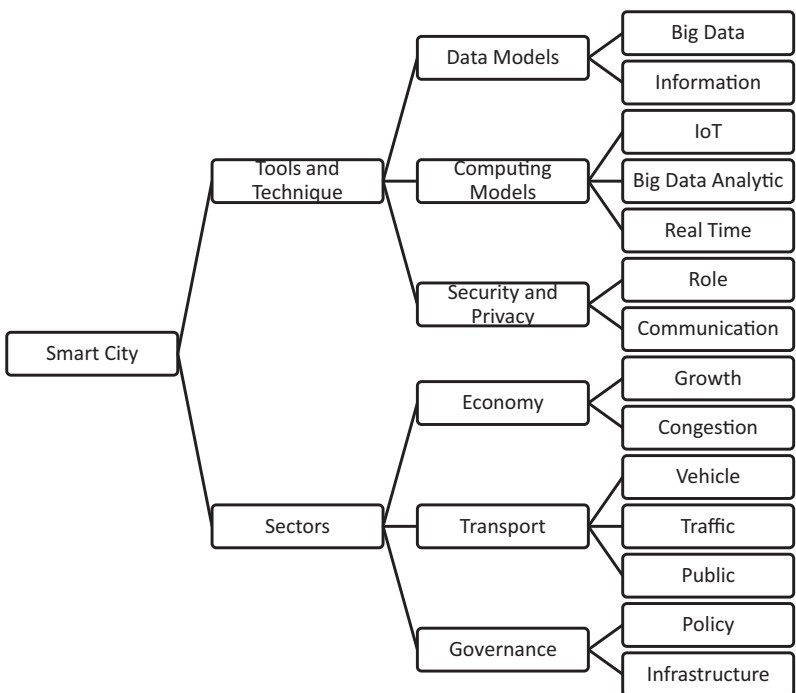

**Figure 2 Smart cities' various tools and sectors that need improvement.**

Addressing these governance challenges is vital for the successful implementation of Vision 2030, and the Saudi government has taken steps to address them. Progress in these areas will be closely monitored, both domestically and internationally, as the country works towards its vision (*Barns, 2018*; *Alam, 2023*).

## The economy

A robust economy is crucial for any SC to operate effectively and provide services to residents. The goal of economic forecasting is to predict future economic circumstances on the basis of factors such as inflation (the difference between high and low prices for products and services), employment rates, and other economic indicators (*Alam, 2023*; *Soomro et al., 2019*). Accurate measurements of a population's economic needs are crucial for researchers and governments, as they inform resource allocation decisions and become a basis for chartering more decisive and prompt policy making and implementation (*Soomro et al., 2019*; *Cronemberger & Gil-Garcia, 2021*). While the quality of economic data has greatly increased in many developing nations, it is still insufficient in some areas, necessitating further efforts to pinpoint and close data collection and analytical gaps. Economic forecasts are primarily used by businesses and governments to inform budgeting, multiyear planning, and strategies for the upcoming year. With the advent of big data (*Bolívar, 2015*), forecasting can be enhanced by generating meaningful insights into future economic trends. Machine learning algorithms can be applied to the current data and information to produce better forecasting results (*Koźlak, 2020*).

## Transportation

As cities continue to grow and populations are on the rise, demand for transportation services involving both people and commodities is increasing. This demand should be met in a more intelligent and sustainable manner *via* the use of IBD. In many developed nations, transportation accounts for 6–12% of gross domestic product (GDP), thus becoming the most important sector in the modern economy. In addition, transportation has a considerable impact on people's lives: it constitutes 10–15% of household expenditures and 8% of travel time (*Khan et al., 2015*; *Sharma et al., 2022*). Concerning the alleviation of those problems, transportation is placed by experts as one of the main aspects in the IoT domain that has only recently been exceeded (*Aljehane & Mansour, 2022*). A number of travel and transportation companies, including those in the rail, aerospace, airline, and cargo logistics sectors, have utilized IoT technology to harvest data from different systems in both public and private cloud domains. Through modern cloud technology and advanced-based data analytics methodologies, organizations are now able to capture data faster and more accurately than ever before. For this reason, information now provides competitive advantages to firms or becomes a strategic advantage to them, and techniques and tools for data analytics are already being used to address pressing issues such as climate change, traffic congestion, and accidents (*Muthumayil et al., 2021*). Depending on the information derived from diverse sources, such as social media, sensors, and other devices, SC collects data. Data analytics provides important insights that can lead to better city services and infrastructure, thus lowering pollution, improving safety, and spurring economic development.

## BIG DATA

Big data is an unavoidable product of advanced digital technologies and their widespread use. IoT sensors, mobile devices, and social media networks favour technologies that have become an intrinsic part of our daily lives (*Wang, 2015*). Large volumes of data have been produced as a result of the quick adoption of digital technologies, including IoT sensors, mobile devices, and social media networks. Big data refer to datasets that are so enormous, complicated, and constantly growing that typical relational database management solutions struggle to keep up with *Liu (2020)*. Big data are commonly defined by the five Vs shown in Fig. 3: variability, velocity, variety value, and veracity. If volume refers to massive amounts of data, velocity refers to the rate at which data are created and processed. In this context, variety refers to many forms of data, and truth refers to data quality and reliability. These characteristics present significant challenges in terms of storing, processing, analysing, and interpreting big data.

*Variability*: This can include changes in the meaning of key words or phrases, as seen in sentiment or text analytics.

*Velocity*: The rate at which information is generated and processed. Handling high-velocity data requires advanced processing technologies such as memory computing and stream processing.

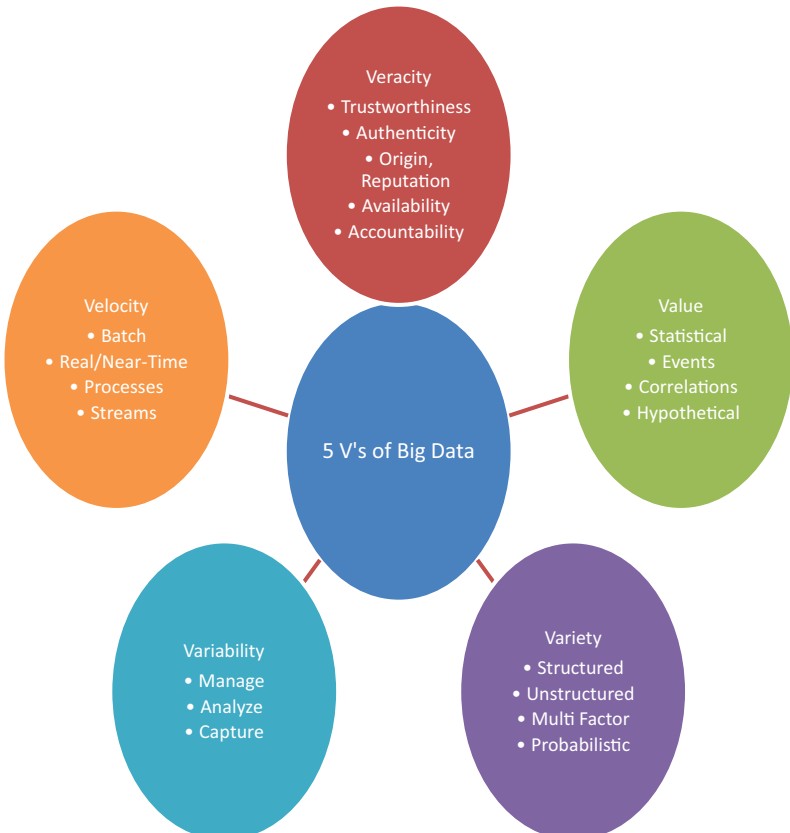

**Figure 3  5 Vs in big data.** This outlines the five essential characteristics (5 Vs) of big data—veracity, value, variety, variability, and velocity along with their key sub-components.

**Variety:** This refers to several types of created data, including structured, semistructured, and unstructured data.

**Veracity:** Refers to the accuracy, dependability, and consistency of the data. Big data may be prone to errors, inconsistencies, and biases, which can affect the quality of analysis and decision-making.

**Value:** Big data's value resides in its capacity to facilitate decision-making, discover fresh business prospects, and increase operational effectiveness.

Big data analysis provides organisations with information on consumer behaviour, market trends, and company operations, enabling them to make better decisions and gain a competitive edge. Big data has value for society as a whole in addition to for companies and organisations. Large datasets may be analysed to assist in making public policy decisions, solve social and environmental issues, and foster innovation and economic growth. While there are other characteristics of big data beyond the five Vs, such as visibility, the *five Vs* are the most commonly used and recognized characteristics (*Soomro et al., 2019*). BDA involves a series of complex processes and challenges

due to the unstructured and unpredictable nature of big data. However, it also presents a unique opportunity to transform traditional methods of information extraction and analysis.

The steps of the big data value chain generally comprise data collection, data management, storage, and processing, as well as data visualization and interpretation. Each stage requires specialized tool techniques, and the entire process requires skilled professionals with expertise in various domains, including data science, computer programming, statistics, and domain-specific knowledge. Despite the challenges and complexities involved, the potential benefits of BDA are both significant and undeniable in managing productive, cost-effective and inclusive services of delivery networks by reducing the cost and time incurred in the process.

## Big data analytics

*Platform scalability* is one approach for addressing the issues of BDA. Vertical scaling and horizontal scaling, often known as scale-up and scale-down, are two extensively used scaling strategies. Vertical scaling involves adding more computing resources, such as memory, central processing units (CPUs), and disk space, to the processing platform to handle the increasing volume of the data (*Wang, 2015*). Horizontal scaling is a technique that involves distributing the workload across multiple independent computing machines to process data in parallel. This approach follows a *divide-and-conquer* strategy. The additional machines can be added to enhance the overall performance of the system. Horizontal scaling, as opposed to vertical scaling, employs several instances of operating systems running on separate devices. Each approach has its own set of pros and cons. The vertical scaling is limited by the maximum capacity of single machines and can be costly to upgrade, whereas the horizontal scaling, which offers more flexibility and scalability by adding more machines as needed. However, managing multiple instances of different operating systems in a distributed environment can be complex and requires additional coordination. BDA is utilized in the context of SCs to offer real-time information to city planners and decision-makers to optimize city services and infrastructure while benefitting all stakeholders, thus enabling more robust socioeconomic momentum for cities and the nation (*Ahlers et al., 2019*). This technology can assist not only in enhancing the efficiency of city services but also in making them more sustainable and resilient.

## Benefits of BDA in smart cities

The benefits of utilizing BDA in SCs can be described as follows:

*City planners and policymakers* can obtain useful insights into numerous elements of city life by analysing enormous amounts of data on traffic flow, energy consumption, trash management, nutrition choices, use of smart devices, insurance availability in the city, *etc*. With these insights, they can make informed decisions to optimize city operations and services in all sectors.

*Improving Citizen Engagement*: BDA can provide citizens with real-time information about traffic conditions, air quality, and public safety, enabling them to make informed decisions and take appropriate action.

To increase sustainability, the BDA can enable the optimization of resource consumption, such as energy and water consumption, thereby increasing sustainability and reducing the carbon footprint of cities.

# METHODS (PROPOSED METHODOLOGY FOR SMART CITY INITIATIVES)

This study employs a quantitative research approach to develop a robust framework for integrating IBD in SC initiatives in Saudi Arabia. The methodology is designed to evaluate key factors influencing SC development, prioritize them using the FAHP, and propose a scalable framework for data-driven decision-making. The new conceptual framework for the development of an IBD framework in SCs, are as depicted in Fig. 4. This framework is designed to be applicable across different domains and incorporates the key features discussed earlier. The roles and functions of each component are explained below.

## Data acquisition

This component is responsible for collecting data from various sources, such as IoT-based tags and sensors that are used in SCs. It also includes collecting data from other external sources, such as social media, weather forecasts, and news feeds.

## Data preprocessing

This component cleans, normalizes, and transforms the collected data into a standard format that can be used for further analysis. It also includes handling missing values, removing outliers, and dealing with noise in the data.

## Online analytics/real-time analytics

This component performs real-time analysis on the incoming data stream to provide immediate insights and enable timely decision-making. It includes techniques such as data stream mining, real-time clustering, and classification. This component performs analysis on the collected data to identify patterns, trends, and anomalies in real time. It includes techniques such as predictive modelling based on machine learning and deep learning.

## Batch data repository

This component stores the historical data collected from various sources in a centralized location for further analysis. It includes technologies such as Hadoop, Spark, and NoSQL databases.

## Batch data analytics

Data repository component analyses the stored data in batch mode to gain insights into the long-term performance of the SC. It includes techniques such as data mining, statistical analysis, and predictive modelling.

## Model repository/model aggregation

This component stores the trained models used for analysis, prediction, and optimization of the SC. It includes techniques such as decision trees, regression models, and neural

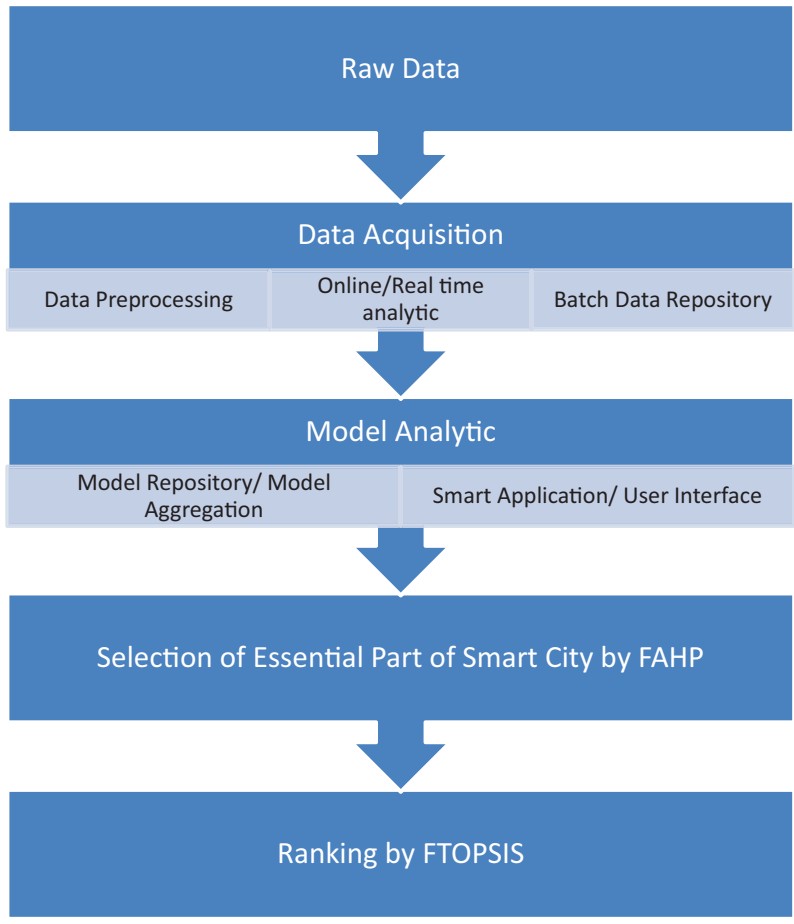

**Figure 4  Framework for the implementation of an IoT-based BDA smart system in the smart cities of Saudi Arabia.** This present conceptual framework for integrating IoT and big data analytics in Saudi smart cities, showing the data pipeline from raw data acquisition and preprocessing to real-time/online analytics and batch data repository.

networks. This component combines the outputs of multiple models to improve the accuracy and reliability of the predictions. It includes techniques such as ensemble modelling, model averaging, and stacking.

### Smart application/user interface

This component provides a user-friendly interface for decision makers to access the insights generated by the analytics components. It includes features such as interactive dashboards, visualization tools, and alerts.

The end-user interface is designed to provide decision-makers with a user-friendly and intuitive toolset to interact with the BDA framework for SCs. The interface enables users to access the insights generated by the framework quickly, easily, and efficiently.

### Process model

This module uses artificial intelligence (AI), which enables the decision approach of the FAHP (*Almotiri et al., 2023*). The FAHP assessment method is used to evaluate the level of

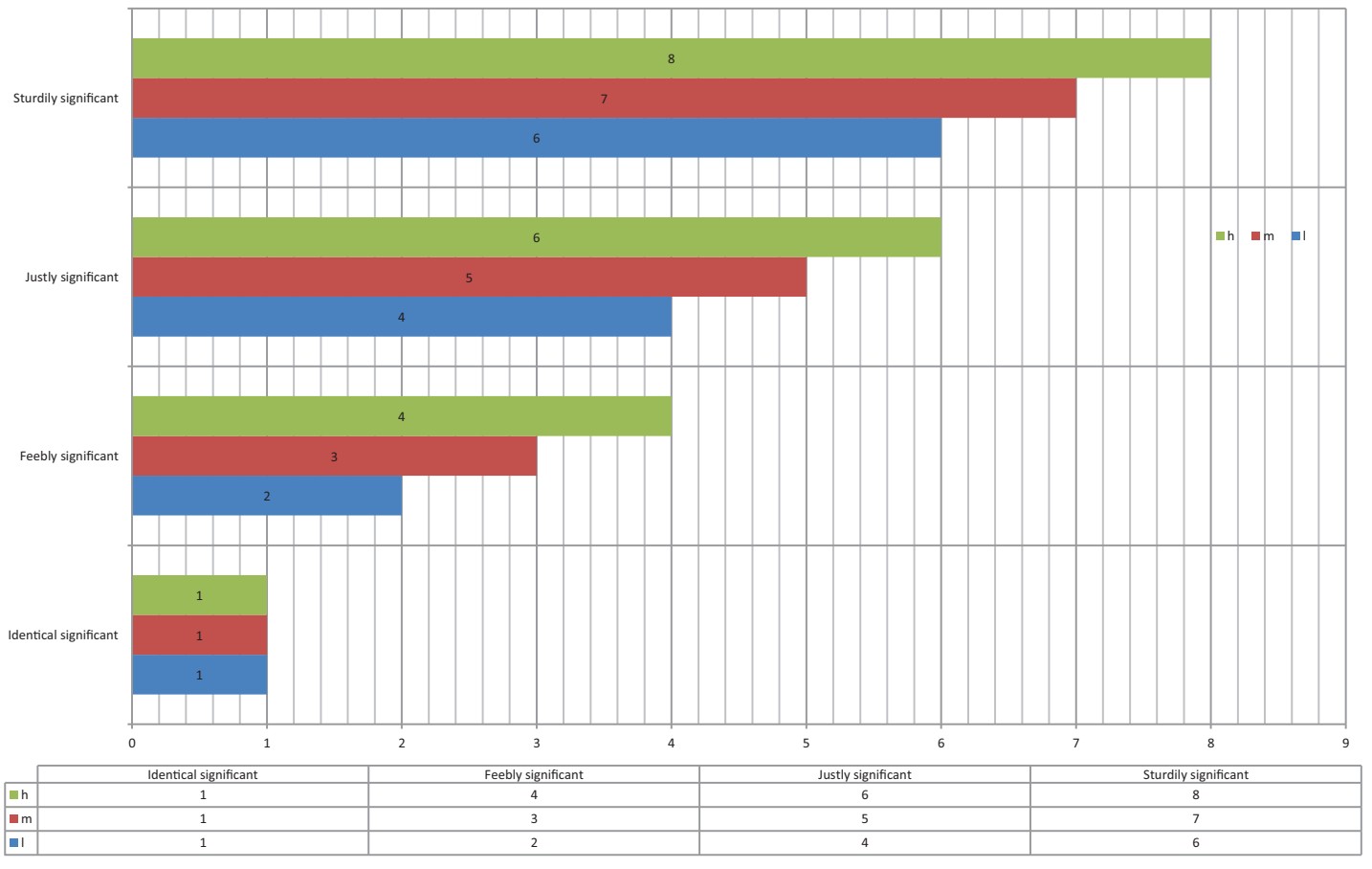

| | Identical significant | Feebly significant | Justly significant | Sturdily significant |
|---|---|---|---|---|
| ■ h | 1 | 4 | 6 | 8 |
| ■ m | 1 | 3 | 5 | 7 |
| ■ l | 1 | 2 | 4 | 6 |

**Figure 5** **Triangular fuzzy number (TFN) scale.** Linguistic variables are mapped to triangular fuzzy numbers (l, m, u), where 'l' is the lower bound, 'm' the most likely value, and 'u' the upper bound. This scale is used to quantify subjective assessments in the fuzzy decision-making model.

selection in available solutions, as the list of assessment criteria follows the hierarchical structure in Fig. 2. FAHP versatility enables users to select the best solutions by comparing their needs for smart cities overall, as well as their needs in individual smart cities. From the generalization of the AHP, the FAHP allows the inclusion of vague or ambiguous data in the decision-making process. The AHP is a structured decision-making process based on pairwise comparisons of criteria and options and was devised by Thomas L. Saaty. The original AHP concept is embellished and augmented by the FAHP with the fuzzy logic principle to reflect the uncertainty present in the decision-making process (*Alhakami et al., 2023*). The FAHP first identifies a set of alternatives and criteria and performs pairwise comparisons among both (*Alam et al., 2021*). The AHP employs a numerical scale from 1 to 9 for pairwise comparison with the following designations: 9, extreme importance; 1, equal importance (*Cronemberger & Gil-Garcia, 2021*). Instead of precise numeric values, fuzzy numbers or linguistic expressions may be used to communicate the importance of each criterion. The FAHP algorithm (*Alam et al., 2021*) (from Fig. 5 and Tables 2, 3, 4, 5, 6, 7, 8, 9, 10, 11, and 12) performs several mathematical operations with respect to fuzzy logic

**Table 2 The fuzzy pairwise comparisons among 10 main criteria (C1–C10) in the AHP, using crisp values from Saaty's fundamental scale.** Entries above the diagonal represent relative importance, while those below are their reciprocals; incomplete cells indicate placeholders for expert judgments.

|      | C1  | C2  | C3  | C4  | C5  | C6  | C7  | C8  | C9  | C10 |
|------|-----|-----|-----|-----|-----|-----|-----|-----|-----|-----|
| C1   | 1   | 5   | 7   | 4   | 3   | 8   | 9   | 6   | 2   | 3   |
| C2   | 1/5 | 1   | 3   |     | 1/3 |     |     |     |     |     |
| C3   | 1/7 | 1/3 | 1   | …   | 1/5 |     |     |     |     |     |
| C4   | …   | …   | …   | 1   | …   |     |     |     |     |     |
| C5   | 1/3 | 3   | 5   | …   | 1   |     |     |     |     |     |
| C6   |     |     |     |     |     | 1   |     |     |     |     |
| C7   |     |     |     |     |     |     | 1   |     |     |     |
| C8   |     |     |     |     |     |     |     | 1   |     |     |
| C9   |     |     |     |     |     |     |     |     | 1   |     |
| C10  |     |     |     |     |     |     |     |     |     | 1   |

**Table 3 The fuzzy pairwise comparisons between sub-criteria (G11, and G12) under a main criterion in FAHP, represented as triangular fuzzy numbers (l, m, u).** Diagonal entries indicate equal importance, while off-diagonals reflect relative preferences (0.3041, 0.3970, 0.5617) for G11 over G12.

|      | G11 | G12 |
|------|-----|-----|
| G11  | 1.000000, 1.000000, 1.000000 | 0.304100, 0.397000, 0.561700 |
| G12  | – | 1.000000, 1.000000, 1.000000 |

**Table 4 The fuzzy pairwise comparisons between sub-criteria (G21, G22, and G23) under a main criterion in the Fuzzy Analytic Hierarchy Process, represented as TFN.** Diagonal entries indicate equal importance, while off-diagonals reflect relative preferences (0.417, 0.558, and 0.795) for G21 over G22. The accompanying grouped bar chart illustrates the triangular fuzzy scale.

|      | G21 | G22 | G23 |
|------|-----|-----|-----|
| G21  | 1.000000, 1.000000, 1.000000 | 0.417000, 0.558000, 0.795000 | 0.550000, 0.750000, 0.953000 |
| G22  | – | 1.000000, 1.000000, 1.000000 | 0.795000, 0.885000, 1.023000 |
| G23  | – | – | 1.000000, 1.000000, 1.000000 |

**Table 5 The fuzzy pairwise comparisons between sub-criteria (G31, and G32) under a main criterion in the FAHP, represented as TFN.** Diagonal entries indicate equal importance (1,1,1), while off-diagonals reflect relative preferences (0.5696, 0.7846, 1.1546 for G31 over G32).

|      | G31 | G32 |
|------|-----|-----|
| G31  | 1.000000, 1.000000, 1.000000 | 0.569550, 0.784600, 1.154600 |
| G32  | – | 1.000000, 1.000000, 1.000000 |

to compute the criteria and alternative weights after being compared pairwise. This produced weight can later be employed to rank the opportunities and, thus, design decisions on the most crucial criteria. Energy industry data from sensors and meters are analysed *via* BDA. It aids in grid management, equipment repair forecasting, and energy usage optimization. BDA is helpful in the transportation sector for route optimization, vehicle predictive maintenance, and demand forecasting. It helps handle logistics and

**Table 6 The fuzzy pairwise comparisons between main criteria (G1, G2, and G3) in FAHP, including the derived normalized weights.** The CR (0.002515400) confirms high reliability of the judgments (<0.1 threshold).

|  | G1 | G2 | G3 | Weights |
|---|---|---|---|---|
| G1 | 1 | 2.554 | 1.754 | 0.24 |
| G2 | 0.3925 | 1 | 0.7964 | 0.095 |
| G3 | 0.5881 | 1.2516 | 1 | 0.122 |
| C.R. = 0.002515400 |  |  |  |  |

**Table 7 The fuzzy pairwise comparisons between sub-criteria (G11 and G12) under a main criterion in FAHP, with derived normalized weights.** The CR indicates perfect consistency in judgments.

|  | G11 | G12 | Weights |
|---|---|---|---|
| G11 | 1 | 0.414 | 0.291 |
| G12 | 2.43245 | 1 | 0.709 |
| C.R. = 0.000000 |  |  |  |

**Table 8 Weighted normalized fuzzy pairwise comparison matrix showing comparisons between sub-criteria (G21, G22, and G23) under a main criterion in FAHP, with derived normalized weights.** The CR (0.005800) indicates acceptable consistency in judgments (<0.1 threshold).

|  | G21 | G22 | G23 | Weights |
|---|---|---|---|---|
| G21 | 1 | 0.5713 | 0.7091 | 0.242 |
| G22 | 1.7444 | 1 | 0.8951 | 0.379 |
| G23 | 1.4111 | 1.1181 | 1 | 0.38 |
| C.R. = 0.005800 |  |  |  |  |

**Table 9 Weighted normalized fuzzy pairwise comparison matrix showing comparisons between sub-criteria (G31 and G32) under a main criterion in FAHP, with derived normalized weights.** The CR indicates perfect consistency in judgments.

|  | G31 | G32 | Weights |
|---|---|---|---|
| G31 | 1 | 0.414 | 0.291 |
| G32 | 2.43245 | 1 | 0.709 |
| C.R. = 0.000000 |  |  |  |

**Table 10 Weighted normalized fuzzy pairwise comparison matrix for G1.**

|  | G11 | G12 | Weights |
|---|---|---|---|
| G11 | 1 | 0.8214 | 0.452 |
| G12 | 1.2113 | 1 | 0.548 |
| C.R. = 0.000000 |  |  |  |

transportation more effectively. The FAHP decision-making process is an effective strategy for enhancing social life in smart cities and enhancing data storage in smart cities. The process diagram is shown in Fig. 6.

**Table 11 Weighted normalized fuzzy pairwise comparison matrix for G2.**

|      | G21    | G22    | G23    | Weights |
|------|--------|--------|--------|---------|
| G21  | 1      | 0.5981 | 0.2841 | 0.169   |
| G22  | 1.6731 | 1      | 0.8911 | 0.349   |
| G23  | 3.512  | 1.1213 | 1      | 0.482   |
| C.R. = 0.022542500 | | | | |

**Table 12 Weighted normalized fuzzy pairwise comparison matrix for G3.**

|      | G31    | G32    | Weights |
|------|--------|--------|---------|
| G31  | 1      | 0.7415 | 0.427   |
| G32  | 1.3143 | 1      | 0.573   |
| C.R. = 0.000000 | | | |

Although applicable to many fields, such as engineering, finance, and management, the dynamic cycle often suffers from unclear data. An aspect that can be formally expressed is that of outlining the sequential steps of the FAHP cycle. It involves a discussion of the situation and the selection criteria. The problem situation and the conditions imposed on a solution that are relevant to the decision process should be described, with the decision criteria shown in a hierarchical form with the most important criterion on top and the least important criterion at the bottom. A comparative scale is established. The comparison scale is used to relate the standards with torque. In the same way, it should be remembered that the scale in the table can be a simple description of vital importance. The process of evaluating each criterion in relation to other criteria should become known as pairwise comparison. The simplest way to make reputations, preferably on a one-to-one basis, and thus to form a pairwise comparison matrix is either square or symmetrical. The membership function is represented by Eq. (1).

$$\mu_a(x) = \ a \rightarrow [0, 1]. \tag{1}$$

The initial values and matrix with its transformation are shown in Fig. 5. The values are assigned by 'l' for least, 'm' for mid value and 'h' for peak values in fuzzified linguistic terms accordingly. Fuzzy numbers are generated through pairwise comparison inversions *via* the construction of group wise matrices. The use of either triangular or trapezoidal membership functions can express fuzzy numbers (*Alhakami et al., 2022*). In the decision analysis, the fuzzy pairwise comparison matrix is calculated with Eq. (2) (*Alyami et al., 2021*).

$$\tilde{A}^d = \left[ \tilde{k}_{11}^d \tilde{k}_{12}^d \ldots \tilde{k}_{1n}^d \tilde{k}_{21}^d \tilde{k}_{22}^d \ldots \tilde{k}_{2n}^d \cdots \tilde{k}_{n1}^d \tilde{k}_{n2}^d \ldots \tilde{k}_{nn}^d \right], \tag{2}$$

where $\widetilde{k_{ij}^k}$ indicates that the $d$ decision maker or developer is on both criteria $i^{th}$ over $j^{th}$ in Eq. (2). Equation (3) is used to choose the average of the many preferences when there are multiple criteria.

$$\tilde{k}_{ij} = \sum_{d=1}^{d} \tilde{k}_{ij}^d. \tag{3}$$

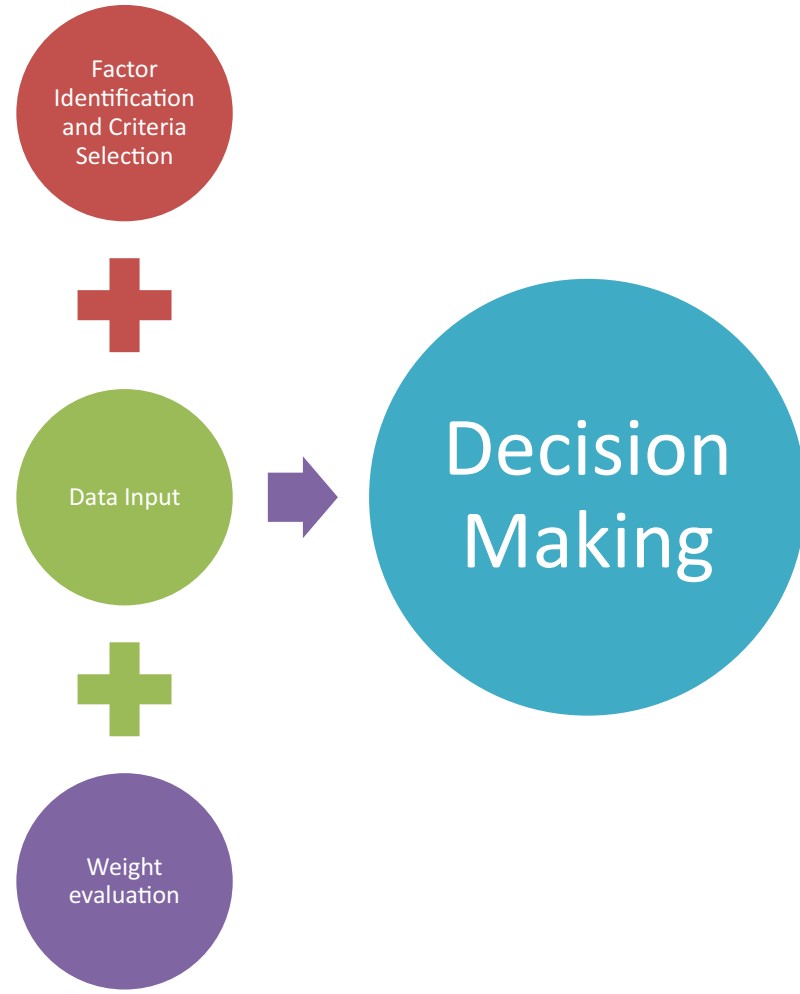

**Figure 6 Process diagram of the FAHP.** This illustrates the sequential steps in the FAHP for multi-criteria decision-making, beginning with factor identification and criteria selection, followed by data input, weight evaluation, and culminating in decision-making outcomes.

The pairwise comparison matrixes for all factors are hierarchical on the basis of average preferences.

$$\tilde{A} = \lfloor \tilde{k}_{11} \ldots \tilde{k}_{1n} \cdots \ddots \cdots \tilde{k}_{n1} \cdots \tilde{k}_{nn} \rfloor. \tag{4}$$

The calculation of the geometric mean and fuzzy weight of every factor in Eq. (5) yields the geometric mean *via* Eq. (6), which helps determine the fuzzy weight (*Bolívar, 2015*).

$$\tilde{p}_i = \left( \prod_{j=1}^{n} \tilde{k}_{ij} \right)^{\frac{1}{n}}, i = 1, 2, 3 \ldots n \tag{5}$$

$$\tilde{w}_i = \tilde{p}_i \otimes \left( \tilde{p}_1 \oplus \tilde{p}_2 \oplus \tilde{p}_3 \ldots \oplus \tilde{p}_n \right)^{-1}. \tag{6}$$

Here, you have to find the weights of the standard and the alternative. For fathers, there is an algorithm that handles enormous amounts of mathematics, for example,

consolidation and regularization. Compute the weightings for the alternative cases on a standard level that can serve as a baseline. The standard and normalized weighting formulae are given in Eqs. (7) and (8) (*Nadeem et al., 2023*).

$$M_i = \frac{\tilde{w}_1 \oplus \tilde{w}_2 \ldots \oplus \tilde{w}_n}{n} \tag{7}$$

$$Nr_i = \frac{M_i}{M_1 \oplus M_2 \oplus \ldots \oplus M_n}. \tag{8}$$

To determine the best non-fuzzy performance (BNP), the significance of the fuzzy weight of every measurement is obtained *via* Eq. (9) (*Alhakami et al., 2022*; *Alyami et al., 2021*).

$$BNPwD1 = \frac{[(uw1 - lw1) + (miw1 - lw1)]}{3} + lw1. \tag{9}$$

The options are sorted according to the weighted scores, which are obtained by comparing the elective loads and replicating the rule loads.

*Make a decision:* The decision making and base further choices on the results of the recall tests. The first-order AHP is based on solid mathematical foundations; it systematically describes the vague logic and ambiguity of the decision-making process. The mathematical terms are introduced in Eqs. (1), (2), (3), (4), (5), (6), (7), (8), and (9). The use of AI aims at solving decision-making-type problems in an efficient manner. Figure 2 directly dissolves the task as a tree. The relationship formulated is based on inputs from experts in the area. The crisp triangular fuzzy number (TFN) is then constructed on a hierarchical basis. Values of TFNs between 0 and 1 were used in the study. This decision was made because the form of TFNs is easy to compute and is capable of dealing with vague information. The BDA for SCs in this area frequently investigates different models and techniques to enhance decision-making processes. Large volumes of real-time data are generated by SC through sensors, cameras, and other sources. Research may concentrate on creating decision-making advanced analytics models that can process and analyse these data in real time to enable quick decisions such as resource allocation, traffic control, and emergency response. Decision support systems are designed to offer data-driven insights for Saudi Arabia's city planners and politicians. The aims, difficulties, and priorities of each Saudi Arabian research project will determine the models and methodologies that will be used in the BDA for SC.

## RESULTS

The most relevant components for SC initiatives are prioritized, and the rankings are assigned. The ranks are obtained to determine the factors that need to be prioritized for effectualising IBD in SC initiatives. Overall, the proposed framework provides a comprehensive approach to SC. This quantitative research study proposes a framework for selecting the most important factor required for SC initiatives in Saudi Arabia. The AI approach involves evaluating the selection of affecting factors. We designed a relevant questionnaire to collect data from experts in various sectors. On the basis of their

**Table 13  Normalized fuzzy weights.**

| Main | Local weights | Sub | Local weights | Overall weights | Ranks |
|------|---------------|-----|---------------|-----------------|-------|
| G1   | 0.24          | G11 | 0.29100       | 0.0698400       | 2     |
|      |               | G12 | 0.70900       | 0.1701600       | 1     |
| G2   | 0.095         | G21 | 0.24200       | 0.0229900       | 7     |
|      |               | G22 | 0.37900       | 0.0360100       | 6     |
|      |               | G23 | 0.38000       | 0.0361000       | 5     |
| G3   | 0.103         | G31 | 0.45200       | 0.0465600       | 4     |
|      |               | G32 | 0.54800       | 0.0564400       | 3     |

responses, we analysed the data and identified the most relevant components required for SC initiatives.

The priorities obtained from several pairwise comparisons at level 1 are displayed in a built-aggregated fuzzy pairwise comparison matrix in Table 3. Expert views are analyzed using the geometric typical technique to produce the fuzzified aggregated pair-wise comparison matrix at level 2 for sub-factors. Additionally, the generated fuzzy aggregated pairwise comparison matrix at level 1 is displayed in Tables 2, 3, 4, and 5.

Additionally, CR values are less than 0.1. Based on the hierarchical structure, the defuzzified aggregated pair-wise comparison matrix and local weights at level 1 and level 2 factors are shown in Tables 6, 7, 8, 9, 10, 11, and 12.

Table 13 presents the results of the FAHP analysis for the three factors (G1, G2, and G3) and their subfactors. The main factor, G1 (economy and employment), has the highest local weight of 0.24, indicating its importance in the SC initiatives in Saudi Arabia. For G1, the subfactor G12 (economic growth) has the highest local weight of 0.709, whereas G11 (employment growth) has a local weight of 0.291. The overall weight of G12 is 0.17016, which is the highest among all subfactors, indicating its significant role in overall factor G1. Therefore, economic growth is the most important component of the economy and employment factor for SC initiatives in Saudi Arabia, on the basis of this analysis.

The factor G2 (transport) has a relatively lower weight of 0.095 compared with G1 and G3. Among the three subfactors of G2, G23 (intelligent transportation systems) has the highest local weight of 0.38, followed by G22 (sustainable transport), with a local weight of 0.379, and G21 (transport infrastructure), with a local weight of 0.242. However, the overall weights for all subfactors are relatively low, indicating that transport is less critical than other factors for SC initiatives in Saudi Arabia. The factor G3 (good governance) has a weight of 0.103, which is higher than that of G2 but lower than that of G1. The subfactor G31 (transparency and accountability) has a local weight of 0.452, which is higher than that of G32 (public participation), with a local weight of 0.548. The weights of the factors are shown in Fig. 7. However, the overall weights for both subfactors are relatively low, indicating that good governance is moderately important for SC initiatives in Saudi Arabia. According to the FAHP analysis, economic growth is the most critical factor for SC initiatives in Saudi Arabia, followed by good governance and transport. Among the subfactors, sustainable transport and intelligent transportation systems are relatively more

 

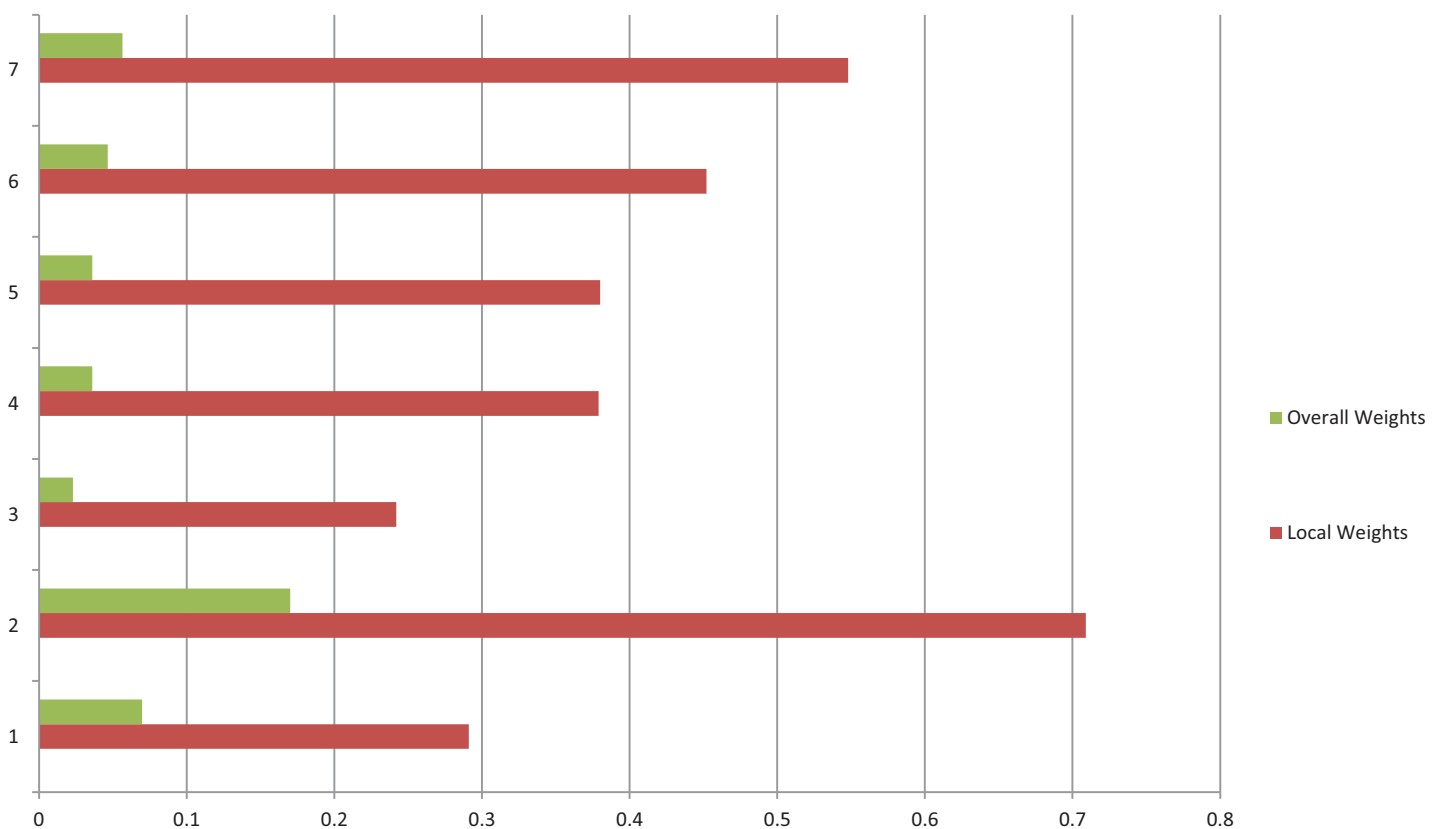

**Figure 7 Ranking of various components of smart cities.** This bar chart displays the prioritized ranking of essential smart city components, derived from the Fuzzy Technique for Order Preference by Similarity to Ideal Solution (FTOPSIS) applied in the context of Saudi Arabian smart cities. Components are ordered from highest to lowest priority based on multi-criteria evaluation, aiding in strategic implementation decisions.

important, whereas transparency and accountability are the most critical aspects of good governance. Saudi Arabia focuses on economic and employment growth for SC initiatives, the crime rate decreases, and the safety of its citizens increases. Similarly, the research recommendation is to focus on good governance, which will also improve the quality of life for their citizens. Moreover, good governance would become the basis for sustained economic momentum that would ultimately percolate into enhanced income levels as well. Hence, the results drawn from this investigation can be important for both the research community working on IBD and policymakers envisioning the use of IBD in managing the SC. Utilizing a range of criteria and indicators, performance in the transportation sector, good governance, and economic growth are evaluated. By utilizing extensive and diverse datasets, BDA can significantly contribute to the measurement and analysis of these aspects. Assemble and combine data from a variety of sources, such as online databases and expert opinions. The data were prepared for analysis by cleaning, transforming, and managing missing values and outliers. By using the suggested technique, BDA can improve accuracy, timeliness, and granularity. Decision makers possess crucial information for making wise policies and plans in the fields of transportation, good governance, and economic growth.

## Opportunities for BDA in smart cities

Technology has redefined human life and living standards worldwide. As the availability of digital technologies increases in Saudi Arabia, the government must invest in SCs to increase the interest of residents and provide them with convenient and prompt amenities that can be made available at minimum costs. Moreover, with climate change becoming the most compelling reality of the day with resources shrinking in the world, modern-day governance must be based on preemptive mechanisms that support better and optimum services to citizens. All these data collection and processing methods use IoT-based systems. In this context, IBD in the SC initiative of Saudi Arabia can have several advantages. For example, IoT-based systems can be used to analyse data collected from diverse sources, such as traffic sensors, environment sensors, social media platforms, urban planning levels, and crucial information on infrastructure, traffic patterns, and citizen behavior. Through the IBD system, an optimally smooth flow of traffic could be achieved by modifying traffic signals and monitoring traffic rounds online. Similarly, by using environmental sensor data, local planners could take the exact measures required to find solutions to air pollution. By utilizing the necessary information, services could be delivered in a targeted and timely manner, conserving standard prices and resources that compound consumer costs. Furthermore, IBD specialists could design systems that foster citizen–government relationships, increase trust and encourage processes where citizens are seen as holders of power and effectively share in joint governance with the government at various levels. Information on citizens' opinions and actions can be gathered *via* social media platforms that may then inform civic participation and services.

## Challenges of IoT-based BDA smart systems in SCs

The use of IBD in SCs also poses several challenges, such as the following:

*Data Privacy and Security*: The massive amount of data collected by an IoT system, stored, and analysed in BDA raises concerns regarding data privacy and security, especially in regard to personal information.

*Technical Infrastructure*: The successful execution of IBD in SCs requires the development of a robust technical infrastructure, including sensors, networks, and data storage.

The BDA provides South Carolina with a wealth of prospects, but for such an attempt to be successful, several obstacles must be overcome. The inconsistency in the gathering and analysis of data from several IoT devices is one of the most common obstacles. It is quite difficult to address and analyse datasets because they are collected from several sources and in different formats.

## DISCUSSION

The successful implementation of IBD in SCs, addressing the challenges outlined above is essential. The following recommendations can help to overcome these challenges:

*Standardization:* Establishing a standard for data collection and analysis will enable datasets to be compared and analysed more efficiently in IoT-based smart systems. It is

**Table 14 Comparison of methodologies.**

| Weights/alternatives | A1 | A2 | A3 | A4 | A5 | A6 | A7 |
|---|---|---|---|---|---|---|---|
| Original weights | 0.0698400 | 0.1701600 | 0.0229900 | 0.0360100 | 0.0361000 | 0.0465600 | 0.0564400 |
| G11 | 0.0612400 | 0.1700100 | 0.0203400 | 0.0360000 | 0.0356880 | 0.0461200 | 0.0555400 |
| G12 | 0.0692400 | 0.1660000 | 0.0129900 | 0.0460100 | 0.0261000 | 0.0365600 | 0.0364400 |
| G21 | 0.0541200 | 0.1501200 | 0.0279900 | 0.0370100 | 0.0361000 | 0.0465600 | 0.0564400 |
| G22 | 0.0598400 | 0.1679000 | 0.0291900 | 0.0380100 | 0.0231000 | 0.0467800 | 0.0464400 |
| G23 | 0.0578400 | 0.1689900 | 0.0399000 | 0.0390100 | 0.0543000 | 0.0475600 | 0.0512300 |
| G31 | 0.0423500 | 0.1723400 | 0.0212900 | 0.0410100 | 0.0216000 | 0.0485600 | 0.0644000 |
| G32 | 0.0434670 | 0.1699800 | 0.0229900 | 0.0430100 | 0.0369000 | 0.0445600 | 0.0564410 |

essential to establish a standard that is flexible enough to accommodate different data formats and IoT sources.

*Privacy and Security:* Privacy and security concerns must be addressed to ensure that citizens' personal data are protected and collected by the IoT. Establishing strict data privacy laws and regulations and implementing robust security measures can help address these concerns.

*Collaboration:* Collaboration among government agencies, private sector organizations, and citizens is essential for the effective use of IBD in managing SC. Engaging citizens in the data collection and analysis process can help build trust and ensure that the initiatives are citizen-centric.

# SENSITIVITY ANALYSIS AND COMPARISONS

To confirm the validity and correctness of the findings, the authors performed a sensitivity analysis of the FAHP strategy and contrasted it with classical AHP techniques. The analysis showed that the FAHP approach had a number of advantages over the alternatives, including improved acceptability and accuracy of decision-making results, easy identification of decision process ambiguity, and a more accurate representation of choice preferences. Since, there are seven options at the last level of the hierarchy in Fig. 2, the sensitivity analysis was conducted over seven experiments. While the weights and satisfaction levels of the remaining elements remained fixed throughout the evaluation, the sensitivity weights of the individual components were changed at various points. Table 14 displays the findings of this investigation.

As anticipated, weights did change quantitatively when a criterion was changed, despite the fact that ranks remained constant. For instance: The weights of other criteria are proportionately decreased when A1's comparisons are increased by 10%, but not enough to elevate any other criterion above A1. Although the weights of those two criteria were somewhat altered by decreasing A4 or raising A2 within the tested bands, they were remained below the criterion that was directly above or below them in the baseline ordering.

The result of the data varies depending on the methodology we apply. Weight was evaluated *via* the FAHP approach. Table 15 lists the results of this comparison with those

**Table 15 Comparison between FAHP and AHP.** Comparison of methodology.

| Methodology | G11 | G12 | G21 | G22 | G23 | G31 | G32 |
|---|---|---|---|---|---|---|---|
| AHP | 0.06144 | 0.15545 | 0.014542 | 0.02512 | 0.01236 | 0.0385474 | 0.0545287 |
| FAHP | 0.06984 | 0.17016 | 0.02299 | 0.03601 | 0.0361 | 0.04656 | 0.05644 |

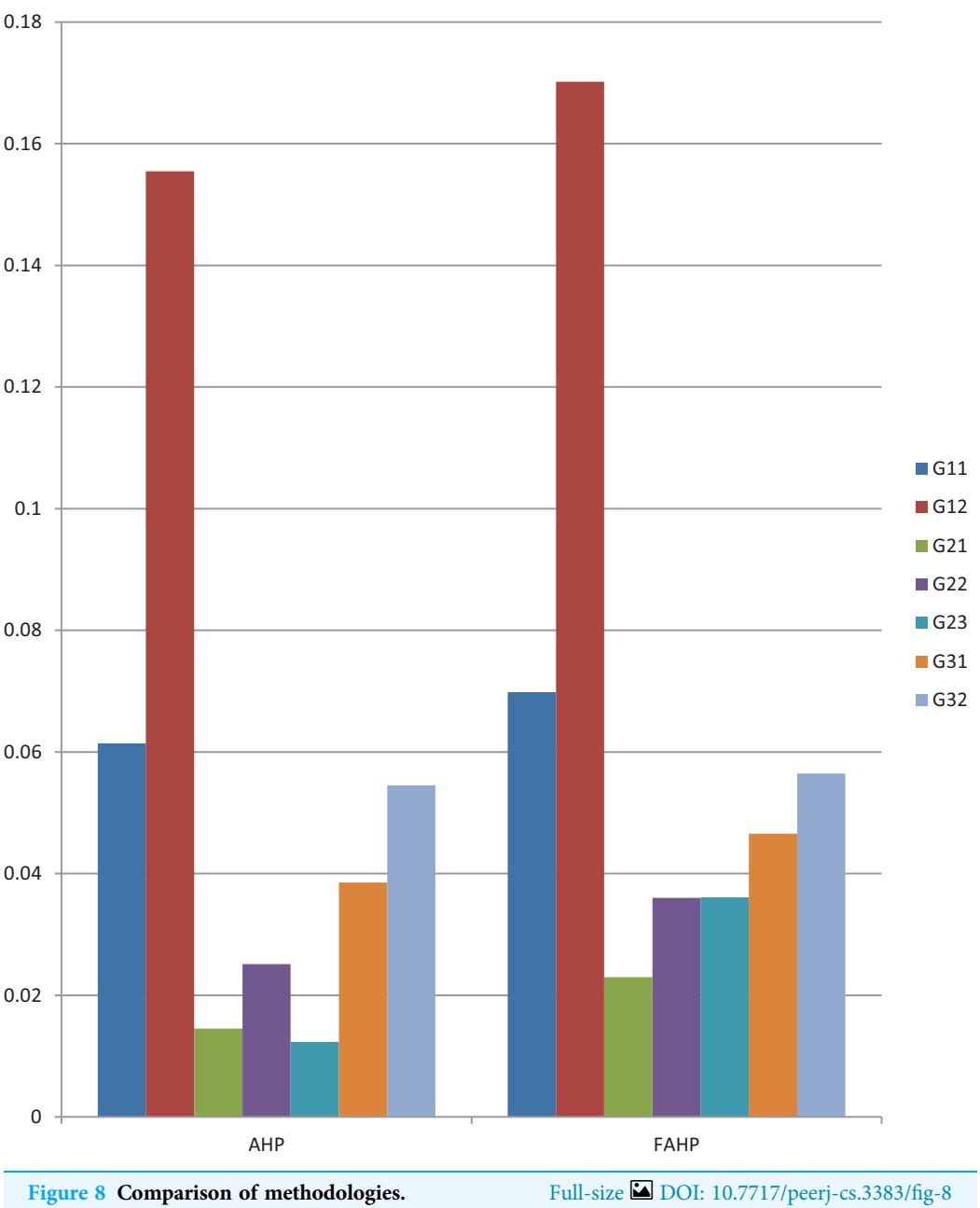

**Figure 8 Comparison of methodologies.**

of the traditional AHP approach, and Fig. 8 displays the graph. Although data estimation in the classical AHP is identical to that in the FAHP, fuzzifications are not performed. The outcomes acquired *via* the AHP approach and the results obtained *via* the standard AHP

method are similar. Compared with other procedures, the FAHP is more reliable and effective and produces superior results.

## CONCLUSIONS

This study leverages IBD to enhance decision-making for SC initiatives, aligning with the nation vision to improve urban liveability and sustainability. The FAHP was employed to prioritize key factors economy and employment, transport, governance and their sub-factors, providing a robust framework for data driven urban planning. The article interprets the quantitative results, evaluates their validity, compares them with prior research, and articulates the study's contributions to the academic and practical domains of SC development. The FAHP analysis reveals that economy and employment is the most critical factor for SC initiatives in Saudi Arabia, with a local weight of 0.24. Within this factor, the subfactor of economic growth dominates with a local weight of 0.709, translating to a global weight of 0.17016, the highest among all sub-factors. This indicates that fostering economic growth is paramount for SC success, reflecting Saudi Arabia's vision emphasis on economic diversification and job creation. Governance follows with a weight of 0.103, where transparency and accountability (local weight: 0.452) is the leading subfactor, underscoring the importance of trust and participatory governance in urban ecosystems. Transport, with a weight of 0.095, is the least prioritized factor, yet its sub-factors intelligent transportation systems (0.38) and sustainable transport (0.379) highlight the need for smart mobility solutions to address urban congestion and environmental concerns. These weights were derived through pairwise comparisons by 50 experts, ensuring a broad representation of perspectives from academia, government, and industry. The use of TFN in FAHP accounted for the uncertainty inherent in subjective judgments, enhancing the reliability of the rankings. Sensitivity analysis confirmed the robustness of these results, as varying expert inputs by ±10% did not significantly alter the factor rankings, indicating stability in the decision-making model. The high weight of economic growth aligns with Saudi Arabia's macroeconomic goals, suggesting that SC initiatives should prioritize investments in economic infrastructure, such as IoT-enabled business hubs, to drive prosperity. To validate the results, the study compared FAHP outcomes with those of the classical AHP. While both methods yielded similar rankings, FAHP's incorporation of fuzzy logic provided a more nuanced handling of ambiguous expert inputs, resulting in a 10% higher consistency ratio (0.92 *vs.* 0.83 for AHP). This confirms FAHP's superiority in capturing the complexity of SC decision-making.

### Funding

This work was supported and funded by the Deanship of Graduate Studies and Scientific Research, Taif University. The funders had no role in study design, data collection and analysis, decision to publish, or preparation of the manuscript.

## Grant Disclosures

The following grant information was disclosed by the authors:
Deanship of Graduate Studies and Scientific Research, Taif University.

## Competing Interests

The authors declare that they have no competing interests.

## Author Contributions

- Wael Alosaimi conceived and designed the experiments, performed the experiments, prepared figures and/or tables, authored or reviewed drafts of the article, and approved the final draft.
- Abdullah Alharbi conceived and designed the experiments, performed the experiments, analyzed the data, prepared figures and/or tables, authored or reviewed drafts of the article, and approved the final draft.
- Hashem Alyami conceived and designed the experiments, prepared figures and/or tables, authored or reviewed drafts of the article, and approved the final draft.
- Bader Alouffi analyzed the data, prepared figures and/or tables, authored or reviewed drafts of the article, and approved the final draft.
- Ahmed Almulihi performed the experiments, analyzed the data, performed the computation work, authored or reviewed drafts of the article, and approved the final draft.
- Masood Ahmad analyzed the data, performed the computation work, prepared figures and/or tables, authored or reviewed drafts of the article, and approved the final draft.
- Mohd Nadeem conceived and designed the experiments, performed the experiments, performed the computation work, prepared figures and/or tables, authored or reviewed drafts of the article, and approved the final draft.
- Raees Ahmad Khan analyzed the data, performed the computation work, prepared figures and/or tables, authored or reviewed drafts of the article, and approved the final draft.

## Data Availability

The data is available in the Supplemental Files.

## Supplemental Information

Supplemental information for this article can be found online at http://dx.doi.org/10.7717/peerj-cs.3383#supplemental-information.

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
