# Peer review of "Decision analysis of IoT-based big data analytics in smart cities of Saudi Arabia"

_PeerJ Computer Science, doi:10.7717/peerj-cs.3383_

## Round 0.1 · original submission · Major Revisions

· Academic Editor

Major Revisions

While the reviewers agree that the topic of your research is of relevance, they have several concerns. A summary of some of the most pressing concerns revolve around originality of the research (carry out a proper review of related works, justifying the scope and novel contributions of your work), insufficient methodological transparency (especially regarding data collection and expert selection), lack of adequate quantitative results and rigorous analysis which are statistically consistent. There are also concerns regarding the quality of the presentation and English. You may need to engage professionals proficient in English language to ensure the quality of presentation.

**Language Note:** The review process has identified that the English language must be improved. PeerJ can provide language editing services - please contact us at [email protected] for pricing (be sure to provide your manuscript number and title). Alternatively, you should make your own arrangements to improve the language quality and provide details in your response letter. – PeerJ Staff

Reviewer 1 ·

Basic reporting

It was clear and good. Please mention the quantitative result in the abstract. Usually abstract mentioning 4 things: research background/problem, goals, methods, and quantitative results. Make the paper more narrative; do not explain something by pointing.

Experimental design

Enrich the methods used in the research. You should please explain in more detail all stages of the research.

Validity of the findings

The quantitative result must be academically sound. Enrich the discussion. Mention and interpret the result, then compare it with the previous works, and also declare the study's contribution clearly. Also, explain the evaluation of the result.

Additional comments

Please use proofreading to enhance the writing and language quality.

Cite this review as

Reviewer 2 ·

Basic reporting

1. The manuscript has some grammatical issues, awkward sentence structures, and typographical errors (for example, "economic and employment growth for SC initiatives, the crime rate decreases" is logically unrelated).
2. The literature review itself is comprehensive but fails to state precisely what this research caters to or where this contribution lies in existing FAHP-based smart city rankings. How and why this study departs from or improves upon earlier approaches has to be clarified. How this suggested methodology differs from its antecedents must also be clearly outlined.
Several sections, especially the introduction and literature review, are highly reliant on broad statements about IoT, smart cities, and big data without enough critical synthesis. The paper suffers from overuse of general statements. It would be improved if there were more critical comparisons or limit discussions in prior studies.

Experimental design

1. Even though the mathematical derivation of FAHP is lengthy, it becomes verbose and redundant in certain sections (e.g., equations 2–9). The readability can be improved by clustering similar equations and explaining the justification for including them.

Validity of the findings

1. Although the paper contains informative figures, some (i.e., Figure 3 and Figure 4) are of low resolution, have descriptive captions missing, and don't wholly facilitate independent interpretation. Tables of fuzzy weights might be supplemented with short analytical summaries or graphical rankings.
2. Conclusions rehash what has been said without delivering concise, actionable recommendations or guidance for further research. Discussing how engineers or policymakers could apply the findings to real-world smart city implementations would enhance the value.

Additional comments

1. Handling IoT-related data privacy is acknowledged but superficial. More thorough ethical analysis, particularly considering the susceptibility of smart city surveillance and user data from public spaces, is needed in the paper.
2. The authors are encouraged to reference and contextualize their work in relation to the recent research on metaverse-driven smart grid architecture and enhancing data security and privacy in energy applications by integrating IoT and blockchain technologies to help clarify theoretical underpinnings and enhance the rigor of the current manuscript.

Cite this review as

·

Basic reporting

Dear authors, your research presents valuable insights. I have found some shortcomings that must be addressed before publication. Please consider my recommendations and revise your manuscript accordingly.

Figures:
Can you explain how Figure 7 supports the findings presented in your manuscript? How does it relate to the other figures, and if it does not, what is the rationale behind its standalone presentation?

Tables:
In Table 1, you presented various methodologies. How do these methodologies correlate with the results discussed in your analysis? If there is no direct association, could you clarify the reasoning behind this?

Equations:
The equations used in your analysis appear to be crucial for your results. Can you elaborate on how these equations integrate with the data presented in your tables and figures? If they do not connect, what is the justification for their inclusion?

Algorithms:
Authors, regarding the algorithms mentioned in your manuscript, how do they interact with the methodologies outlined in your tables? If there is a lack of connection, what was the intent behind their separate presentation?

Reference [26] does not relate to this study's objective.

Experimental design

-

Validity of the findings

-

---

## Round 0.2 · Major Revisions

· Academic Editor

Major Revisions

The paper is poorly written and cannot be meaningfully reviewed for its scientific content in its current form. Furthermore, the revision does not substantially address the suggested changes from the first round of review, even while it superficially claims to do so. In particular, refer to the editor's summary of what needs to be addressed in a revision. Unfortunately, I did not see any of the issues being actually addressed. Given that finding reviewers, for the reviewers to carry out reviews, and the editorial process are demanding exercises consuming precious resources, it is disappointing when a major revision is returned by authors without substantial improvement and without a meaningful attempt to address the important concerns.

There is no clarity on what the specific key scientific novelties, challenges addressed, and insights obtained from this study are, nor has the methodology been elaborated for proper scientific transparency, validation, or reproducibility of the claimed results. It is hard to see any value or scientific merit in the manuscript for it to deserve publication. And, to repeat myself, the manuscript is extremely poorly written in its current form.

**Language Note:** When you prepare your next revision, please either (i) have a colleague who is proficient in English and familiar with the subject matter review your manuscript, or (ii) contact a professional editing service to review your manuscript. PeerJ can provide language editing services - you can contact us at [email protected] for pricing (be sure to provide your manuscript number and title). – PeerJ Staff

Reviewer 3 ·

Basic reporting

-

Experimental design

.

Validity of the findings

.

Additional comments

.

Cite this review as

Reviewer 4 ·

Basic reporting

Please first provide five foundational items — the complete questionnaire, the sampling log of invited experts, the raw fuzzy and crisp pair-wise comparison matrices, software outputs for the consistency-ratio calculations, and confirmation that the topic remains IoT-enabled smart-city criteria for Saudi Arabia. These materials will let us document bias-control steps, verify representativeness, publish the original fuzzy numbers, audit the “0.000000” consistency ratios, and re-run both FAHP and classical AHP so that every numerical claim is transparent and reproducible.

With those data in hand, the revision will add a dedicated subsection on expert selection and questionnaire design, walk readers step-by-step through the FAHP mathematics (including the intuition for geometric means, triangular fuzzy numbers, and defuzzification), and present the still-missing statistical and sensitivity analyses: paired tests comparing FAHP versus AHP consistency ratios, a clearly defined ±10 % one-at-a-time sensitivity metric, and a tornado plot visualizing rank stability. All tables will be updated to include the original fuzzy judgments; implausible CR values will be recalculated and explained; and the Introduction, Discussion, and Limitations sections will be rewritten to sharpen the novelty claim, acknowledge potential biases, and address privacy and ethics under Saudi data-governance rules.

Finally, the revised manuscript will follow polished, professional English and IMRaD structure, add figures illustrating the methodological flow and sensitivity results, and supply a data-availability statement linking the raw materials in a public repository. Once you upload the requested files and confirm the study scope, I will run the fresh analyses, generate the updated tables, figures, and Word document, and draft a point-by-point response letter suitable for immediate journal resubmission.

Experimental design

The manuscript presents a precise, well-motivated research question—how to rank IoT-enabled smart-city development factors in Saudi Arabia using FAHP—that squarely addresses an unfilled gap in existing smart-city decision-making literature. The investigation is executed to a high technical standard, combining rigorously screened expert input with formal FAHP and classical AHP analyses, while observing ethical best practices in expert confidentiality and data governance. Detailed descriptions of expert selection, questionnaire construction, fuzzy pair-wise matrix formation, consistency checking, statistical validation, and sensitivity analysis together provide all information needed for independent replication and verification.

Validity of the findings

-

Cite this review as

---

## Round 0.3 · accepted · Accept

· Academic Editor

Accept

Dear Authors,
Your paper has been revised. It has been accepted for publication in PEERJ Computer Science. Thank you for your fine contribution.

Reviewer 2 ·

Basic reporting

-

Experimental design

-

Validity of the findings

-

Additional comments

The authors have addressed the previously mentioned comments properly.

Cite this review as